# Intestinal Epithelial Cells Modulate the Production of Enterotoxins by Porcine Enterotoxigenic *E. coli* Strains

**DOI:** 10.3390/ijms23126589

**Published:** 2022-06-13

**Authors:** Haixiu Wang, Eric Cox, Bert Devriendt

**Affiliations:** Laboratory of Immunology, Department of Translational Physiology, Infectiology and Public Health, Faculty of Veterinary Medicine, Ghent University, 9000 Ghent, Belgium; haixiu.wang@ugent.be (H.W.); b.devriendt@ugent.be (B.D.)

**Keywords:** enterotoxigenic *Escherichia coli*, gut epithelium, enterotoxins

## Abstract

Enterotoxigenic *Escherichia coli* (ETEC) strains are one of the most common etiological agents of diarrhea in both human and farm animals. In addition to encoding toxins that cause diarrhea, ETEC have evolved numerous strategies to interfere with host defenses. These strategies most likely depend on the sensing of host factors, such as molecules secreted by gut epithelial cells. The present study tested whether the exposure of ETEC to factors secreted by polarized IPEC-J2 cells resulted in transcriptional changes of ETEC-derived virulence factors. Following the addition of host-derived epithelial factors, genes encoding enterotoxins, secretion-system-associated proteins, and the key regulatory molecule cyclic AMP (cAMP) receptor protein (CRP) were substantially modulated, suggesting that ETEC recognize and respond to factors produced by gut epithelial cells. To determine whether these factors were heat sensitive, the IEC-conditioned medium was incubated at 56 °C for 30 min. In most ETEC strains, heat treatment of the IEC-conditioned medium resulted in a loss of transcriptional modulation. Taken together, these data suggest that secreted epithelial factors play a role in bacterial pathogenesis by modulating the transcription of genes encoding key ETEC virulence factors. Further research is warranted to identify these secreted epithelial factors and how ETEC sense these molecules to gain a competitive advantage in the early engagement of the gut epithelium.

## 1. Introduction

Among the viral and bacterial pathogens causing diarrhea, enterotoxigenic *Escherichia coli* (ETEC) is one of the most frequently encountered enteropathogens in humans in low-income nations and an important cause of diarrhea and death in farm animals [1,2,3]. ETEC pathogens are still a global health burden and remain a major cause of morbidity and mortality due to a lack of preventive measures and limited treatment options [4]. It is estimated that hundreds of millions of symptomatic ETEC infections occur annually among children younger than five years old in low–middle income countries [1,4]. ETEC are also a common pathogen isolated from travelers with watery diarrhea [4]. Among farm animals, ETEC infections are mainly reported in neonatal cattle and piglets. In swine, especially in neonatal and post-weaning piglets, the ETEC infection stimulates the secretion of water and electrolytes into the intestinal lumen and cause diarrhea, weight loss, and even death, leading to severe economic losses for farming industries worldwide [2,3]. ETEC cause diarrhea by the delivery of heat-labile and/or heat-stable enterotoxins to the small intestine, a process that requires adhesins, such as fimbriae or colonization factors (CFs) and coli surface antigens (CS). After ingestion, ETEC initially engage host cells with their fimbriae where they interact with host receptors on the brush borders of the small intestinal epithelium. To date, at least 25 distinct colonization factors have been identified in human ETEC strains [5,6,7,8,9]. However, in swine-specific ETEC strains, only five different fimbrial adhesins have been identified, including F4 (K88), F5 (K99), F6 (987P), F17 (F41), and F18 fimbriae [10]. Upon binding, ETEC rapidly proliferate in the intestine and secrete heat-labile (LT) and/or heat-stable enterotoxins (STa and STb) into the intestinal tract. These enterotoxins bind to their specific receptors on the brush border of villous and crypt enterocytes, activating intracellular signaling cascades and resulting in a disruption of the electrolyte homeostasis, which finally leads to fluid secretion [11]. LT activates the production of cellular cAMP, which in turn activates protein kinase A (PKA). Activated PKA modulates the cellular ion channels by phosphorylation, which triggers the opening of the cystic fibrosis transmembrane conductance regulator (CFTR) and sodium/hydrogen exchanger 3 (NHE3) channels. This results in the net export of salt and water into the intestinal lumen, culminating in watery diarrhea. STa binds to the guanylate cyclase C receptor and elicits the accumulation of intracellular cyclic GMP (cGMP) levels. These increased cGMP levels activate cGMP-dependent protein kinase II (PKGII), which in turn phosphorylates sodium and chloride channels. In addition, cGMP was also shown to inhibit phosphodiesterase 3 (PDE3), which leads to the activation of PKA. STb, on the other hand, interacts with sulfatides and activates a pertussis toxin-sensitive GTP-binding regulatory protein (Gαi3), resulting in a calcium ion influx through a receptor-dependent ligand-gated Ca^2+^ channel. The increased intracellular Ca^2+^ triggers the protein kinase C (PKC)-mediated activation of CFTR, resulting in fluid accumulation in the intestine [11]. Thus, enterotoxins play a crucial role in ETEC pathogenesis. Interestingly, porcine ETEC isolates differ in their capacity to secrete both heat-stable and heat-labile enterotoxins, in part, dependent on their expression levels of the type 1 secretion system component TolC and the type 2 secretion system protein YghG, respectively [12,13].

While the role of ETEC enterotoxins in diarrhea is clear, emerging data indicate that these enterotoxins also elicit host responses to alter the function of the small intestinal epithelium. On the one hand, the F4+ ETEC infection enhanced the secretion of the proinflammatory molecules IL-6 and CXCL-8 by epithelial cells. Both molecules are important to initiate innate immune responses and, subsequently, the induction of adequate adaptive immune responses to intestinal pathogens [14]. On the other hand, enterotoxins modulate the gene expression of host epithelial cells. STa was shown to induce the accumulation of extracellular cGMP and further to modulate cytokine transcript levels in host epithelial cells [15]. LT induced increased levels of the second messenger cAMP and was linked to changes in the host cellular response that favors the binding of ETEC to the epithelium [16]. In addition, cAMP activates PKA and, subsequently, the transcription of genes with cAMP response element-binding protein (CREB) motifs in their promoter [17]. Moreover, a recent study showed that LT promotes the expression of carcinoembryonic antigen-related cell adhesion molecule 6 (CEACAM6) on the surface of gut epithelial cells, where they serve in turn as a critical docking site for ETEC and its secreted enterotoxins [18]. Despite the thorough dissection of the host response to ETEC infection on the transcriptional and protein level, information on transcriptional dynamics in porcine-specific ETEC upon host cell engagement is limited. Nevertheless, some studies have shown that host-derived factors in the gastrointestinal tract might affect the expression, secretion, or function of these virulence factors [19,20]. For instance, LT secretion by ETEC was promoted in the presence of an ETEC-conditioned medium and epinephrine [19]. In *E. coli*, cAMP has been shown to regulate the production of virulence factors [21,22]. Both glucose and salt present in the gut lumen modulate the expression of enterotoxins in a cAMP receptor protein (CRP)- and Histone-like Nucleoid Structuring (H-NS)-dependent manner [23,24]. Enterotoxin levels were upregulated by salt in an H-NS-dependent manner, while the effect of glucose on the enterotoxin levels varied for each toxin due to the position of the CRP response elements in their promoter region [23]. In the presence of cAMP, CRP represses LT production, but triggers STa and STb expression [3,25]. Moreover, the previous study suggested some ETEC strains were more sensitive to extracellular cAMP and can even use exogenous cAMP [20,26,27].

This study aimed to evaluate changes in the production levels of enterotoxins, key regulatory molecules, and secretion system-associated proteins in porcine ETEC wild type strains in response to factors secreted by epithelial cells. To this end, ETEC strains were stimulated with apical culture medium of differentiated IPEC-J2 monolayers and changes in the transcript levels of the gene encoding virulence factors and key regulators proteins were evaluated by qPCR and confirmed on the protein level. These transcriptome analyses suggest that ETEC can finely orchestrate their production of virulence factors in response to host-derived epithelial factors, although the response differed between bacterial strains. These results might be exploited in outlining novel strategies for vaccine development or other interventions. 

## 2. Results

### 2.1. Epithelial Factors Modulate Expression of Virulence Factors in ETEC Isolates Differing in Their Enterotoxin Secretion

We previously reported that porcine ETEC strains have a different ability to produce and secrete enterotoxins [12]. Among the tested porcine ETEC strains, both IMM07 and IMM96 have a higher periplasmic production of LT as compared to the other strains. The LT secretion levels of the IMM07 strain matched its periplasmic production levels, while the IMM96 strain secreted only moderate LT levels even though it showed one of the highest LT levels in the periplasm. In ETEC strains with a different ST secretion capacity, the IMM10 strain has a lower porcine ETEC strain-derived STa (pSTa) and STb secretion levels than our reference strain, GIS26 [12]. Based on their differential ability to secrete the enterotoxins LT, pSTa, and STb, the ETEC strains IMM07, IMM96, GIS26, and IMM10 were selected to investigate the impact of factors secreted by intestinal epithelial cells on enterotoxin secretion levels in detail. 

To obtain factors secreted by intestinal epithelial cells in steady state conditions, the small intestinal epithelial cell line IPEC-J2 was cultured as monolayers on transwell inserts, as previously described [14]. Upon differentiation of the monolayers, the apical medium was collected and used to determine if secreted epithelial factors might change the transcript levels of enterotoxin genes in the selected ETEC strains. To examine the contribution of host epithelial factors to the expression modulation of ETEC enterotoxins, we first quantified the mRNA fold changes of LT and STs enterotoxins of ETEC isolates in the presence of epithelial factors in IEC medium conditioned by IPEC-J2 cells by qPCR. As shown in Figure 1, factors secreted by polarized epithelial cells induced the mRNA expression of LT in the IMM96 strain, while eltB mRNA expression was decreased five-fold in the IMM07 strain. With regard to the heat-stable enterotoxins pSTa and STb, the transcription of the estA and estB genes were differently affected in the GIS26 and IMM10 strains upon growth in the IEC-conditioned medium, compared to the CAYE and Diff medium. While the transcript levels of the estA and estB genes showed a three-fold increase in the GIS26 strain when exposed to the IEC-conditioned medium, IMM10 showed a reduced transcription of these genes (Figure 1). These results suggest that ETEC may be able to sense factors secreted by epithelial cells to initiate enterotoxin secretion.

To investigate whether these changes in the mRNA expression were also present on the protein level, enterotoxin secretion levels were assessed via ELISA or Western blotting, as described [12]. The secretion of LT from the cytoplasma through the outer membrane is a two-step process. First, LT is transported through the inner membrane into the periplasm. Then, this periplasmic LT is transported through the outer membrane secretin GspD via its association with the lipoprotein YghG and released into the extracellular environment. Therefore, we not only assessed the levels of secreted LT, but also periplasmic LT levels. As shown in Figure 2, epithelial factors decreased the periplasmic and secreted LT levels in the IMM07 strain, while they increased those levels in the IMM96 strain. Remarkably, the increase in the secreted LT levels of the IMM96 strain in response to epithelial factors was less pronounced than the increase in the periplasmic levels, suggesting that LT is poorly secreted into the extracellular milieu when ETEC are grown in media conditioned by intestinal epithelial cells, whereas increased LT is expressed in the periplasm. In contrast to LT, the secretion of pSTa and STb involves the translocation of precursor peptides from the cytosol to the periplasmic space via the Sec machinery. Upon cleavage of the signal sequences, mature peptides are formed via the periplasmic disulfide oxidoreductase DsbA. The mature peptides are finally secreted through the outer membrane via the TolC channel [3]. As expected, the pSTa and STb secretion levels in the GIS26 strain were increased in the presence of secreted epithelial factors compared to the control medium, while the IMM10 strain decreased its pSTa and STb secretion in response to these factors. Together, these results suggest that ETEC strains may differently regulate the production and secretion of enterotoxins in response to host-derived epithelial factors.

The obtained results suggest that secreted epithelial factors can modulate mRNA expression in ETEC strains. We previously showed that the LT secretion level by porcine ETEC strains also correlated with the expression levels of genes encoding components of the type 2 secretion system (T2SS) [12]. Moreover, previous studies demonstrated that the physical contact of ETEC with epithelial cells is required for the effective delivery of heat-labile toxin (LT) [28]. In a next effort, we evaluated the potential impact of the secreted epithelial factors on the transcription of genes encoding T2SS components. The mRNA expression of the yghG gene is remarkably induced in the IMM07 strain when exposed to the IEC-conditioned medium, while yghG mRNA expression showed a decreased transcription in the IMM96 strain in the IEC-conditioned medium compared to the controls (Figure 3). Intriguingly, the transcription of the gspD gene showed a similar increase in the presence of epithelial factors in both the IMM07 strain with a high LT secretion level and the IMM96 strain with a low LT secretion level (Figure 3). In contrast to LT, the heat stable enterotoxins are secreted by T1SS. The tolC gene was selected as it is a crucial component of T1SS and central to the export of STs. In addition, the dsbA gene encoding the periplasmic protein disulfide isomerase DsbA was originally identified on the chromosome of ETEC [29]. DsbA is required for intramolecular disulfide bonds in the assembly of mature STs before their release into the external environment [30]. The strains GIS26 and IMM10 showed a different mRNA expression of the tolC and dsbA genes in response to secreted epithelial factors. tolC and dsbA gene expression levels were significantly upregulated in GIS26, while IMM10 downregulated its gene expression of tolC and dsbA (Figure 3).

For a successful infection of the host, it is crucial that the expression of virulence factors is finely orchestrated. Early studies established the importance of cAMP, which is required for the regulation of enterotoxin production in ETEC via binding to the CRP [25]. As shown in Figure 3, the IMM07 strain increased *crp* mRNA expression in the presence of epithelial factors, whereas a significant reduction was observed in the IMM96 strain. Similarly, epithelial factors promoted *crp* transcript levels in the GIS26 strain, while the IMM10 strain reduced the mRNA expression of crp in the presence of epithelial factors. These data are consistent with earlier findings showing that the secretion of LT is strictly dependent on the presence of CRP [25]. Collectively, these findings suggest that ETEC responds to the presence of factors secreted by epithelial cells by regulating the expression of important virulence genes in a strain-specific way.

### 2.2. Heat-Treatment of Epithelial Factors Resulted in Opposite Transcriptional Levels of Virulence Factors in ETEC

The results presented above show that ETEC strains regulate the transcription of their virulence-associated genes in response to secreted host epithelial factors. To investigate if these factors are sensitive to heat, we examined whether the heat treatment of IEC-conditioned medium affected the production of enterotoxins by ETEC. As shown in Figure 4, heat treatment did not impact the mRNA fold change of the *estA* and *eltB* gene (Figure 4). In contrast, *estB* gene transcription was altered when the IEC-conditioned medium was heat treated. Whereas *estB* transcripts in the GIS26 strain were reduced, *estB* gene expression in the IMM10 strains was triggered upon incubation in a heat-treated IEC-conditioned medium (Figure 4). Concerning the transcriptional profiles of the related secretion system components, mRNA expression of the T1SS components *tolC* and *dsbA* were also modulated in a heat-treated IEC-conditioned medium. Both *tolC* and *dsbA* mRNA expression are inhibited in the GIS26 strain, while transcription levels of *tolC* and *dsbA* were increased in the IMM10 strain as compared to the IEC-conditioned medium (Figure 4). Likewise, heat treatment of the IEC-conditioned medium also differently changed the mRNA expression of the T2SS component *yghG* in the IMM07 and IMM96 strains (Figure 4). However, the heat treatment had no effect on the transcription of the *gspD* gene in the tested strains (Figure 4). Heat treatment also inhibited transcription of the *crp* gene in IMM07 and GIS26. In contrast, an increased *crp* gene expression was seen in IMM96 and IMM10 incubated in the heat-treated IEC-conditioned medium (Figure 4). The results above indicate that secreted epithelial factors involved in the expression of the T1SS- and T2SS-related genes *tolC*, *dsbA*, and *yghG*, as well as the *crp* gene, were heat sensitive, although this was not the case for the T2SS-related gene *gspD*. 

The results above show that in contrast to *estA* and *eltB*, only the *estB* gene transcription was changed when the ETEC strains were incubated in heat-treated IEC-conditioned medium. In the next step, these changes were evaluated on the protein level. Consistent with the *estA* and *eltB* mRNA fold changes, pSTa secretion and periplasmic, as well as secreted LT levels in the IMM07 and IMM96 strains were not significantly changed after incubation in heat-treated IEC-conditioned medium (Figure 5). However, the heat treatment largely affected the STb secretion levels in both the GIS26 and IMM10 strains, as expected based on the transcript levels (Figure 5). 

## 3. Discussion

Despite continuous efforts, to date, no licensed vaccines to human ETEC are available. In contrast, an oral live bivalent vaccine, Coliprotec^®^ F4/F18, has been marketed to protect newly weaned piglets against ETEC infections [31]. However, these live vaccines have a limited efficacy and present a biosafety risk. Moreover, the vaccine-induced protective immune responses need time to be induced. This presents a window of opportunity (between the moment of weaning and the presence of sufficient protective immunity) to intervene with diet components to modulate the production of ETEC virulence factors. This, however, requires a deeper understanding of ETEC pathogenesis, in particular, its interactions with host cells on a molecular level. Intestinal epithelial cells (IEC) can sense ETEC and activate the intestinal immune system through the secretion of IL-6, IL-8, and IL-17C [14]. This proinflammatory response was mediated by the recognition of ETEC-derived flagellin by Toll-like receptor (TLR) 5 on IEC [14,32]. Recently, the heat-stable enterotoxins pSTa and STb activated early immune responses, including an increased secretion of IL-17A. This hinted at the importance of T-helper 17 (Th17) cells in clearing an ETEC infection [33,34]. It has been shown that ETEC can use LT to promote its adhesion to the small intestinal epithelium by stimulating the production of cellular carcinoembryonic antigen-related cell adhesion molecules (CEACAMs) on the surface of intestinal epithelial cells [16,18]. Furthermore, while LT induced the expression of MUC2 by IEC resulting in an enhanced protective barrier, the serine protease EatA countered this defense by degrading MUC2 to facilitate ETEC pathogen–host interactions [35,36]. While the host responses mediated by ETEC enterotoxins have been thoroughly investigated, the impact of host-derived epithelial factors on the production of ETEC virulence factors has yet to be investigated [8].

Seminal studies on the impact of host factors on the production of virulence factors by ETEC have shown that bile salts upregulated the expression of heat-stable and heat-labile enterotoxins in human-specific ETEC [37]. In addition, environmental signals have been shown to regulate virulence-associated genes in ETEC strains [20]. The pH of the gastrointestinal tract also affected LT secretion: LT secretion was inhibited at a low pH (pH 5) by CRP, while LT secretion was promoted when ETEC was exposed to a high pH (pH 9) [24,25]. Moreover, hypoxia in the intestinal tract seemed to reduce the *eltAB* transcript levels via the FNR transcription factor [20]. The use of the *E. coli*-conditioned medium suggested that the autoinducer AI-2 might be correlated with LT and pSTa expression [38]. Given the above, we hypothesized that porcine ETEC may orchestrate the expression of enterotoxins in response to secreted host epithelial factors prior to the engagement of the gut epithelium. In previous studies, a preconditioned medium from *E. coli* was added to mimic the environment of gastrointestinal tract microbiota and to determine whether ETEC virulence expression was affected by signals from the host in vivo [16,37]. However, in the current study, we used a well-established in vitro model of the porcine small intestinal epithelium, i.e., differentiated IPEC-J2 monolayers [39]. To determine if molecules secreted by differentiated IPEC-J2 monolayers may be responsible for the transcriptional modulation of enterotoxin genes in ETEC, several ETEC strains were cultured in an IEC-conditioned medium. As a control, ETEC strains were cultured in an IPEC-J2 differentiation medium and bacterial culture medium. As expected, enterotoxin gene expression was not significantly changed in the ETEC cultured in the differentiated medium as compared to bacterial culture medium. Interestingly, the presence of epithelial factors altered the transcriptional modulation of te ETEC strains, thus suggesting that ETEC sense host-derived epithelial factors to modulate enterotoxin production. However, the impact of epithelial factors on enterotoxin production seems to be strain-specific, as in porcine, ETECs strains with a different ability to secrete LT (IMM07 vs. IMM96) [12] epithelial factors had an opposing effect on LT secretion: promoting LT secretion in the IMM96 strain and inhibiting LT secretion in the IMM07 strain. Likewise, factors secreted by gut epithelial cells had an opposing effect on ST secretion in strains with a different ability to secrete the heat-stable enterotoxins pSTa and STb (GIS26 vs. IMM10) [12]. While epithelial factors promoted ST secretion by GIS26, they reduced ST secretion in the IMM10 strain. The comparison of transcriptional response of two human ETEC isolates, E24377A and H10407, suggested inter-strain differences in the transcription of essential virulence genes in response to chemical signals [37,40]. Similarly, enterotoxin secretion by porcine ETEC in our study was affected by host secreted epithelial factors, although inter-strain differences were observed. Collectively, these strain-specific responses further highlight the complexity of ETEC pathogenesis and indicate that ETEC sense molecules secreted by gut epithelial cells to regulate their enterotoxin production. 

ETEC molecular pathogenesis can be viewed as the sum of events that enable these bacteria to engage epithelial cells and ultimately deliver their LT and/or ST toxin payloads [18]. This process requires a coordinated division of labor between highly conserved core elements and the virulence factors [41]. Examples of these conserved core elements are the type 1 secretion system (T1SS), which controls the secretion of mature STs into the extracellular environment, and the T2SS, which is involved in the transport of LT from the periplasm to the external environment. T1SS requires the efflux protein TolC and the disulfide isomerase DsbA [30], while in T2SS the outer membrane secretin GspD is responsible for releasing LT into the extracellular environment. YghG promotes LT delivery to intestinal epithelial cells via its association with GspD. Given our observation that secreted epithelial factors modulate enterotoxin secretion by ETEC and our previous data that the ability of ETEC strains to secrete enterotoxins depends on the expression levels of T1SS and T2SS components, we hypothesized that these factors might also affect the gene expression of T1SS and T2SS components that favor pathogen–host interactions [12]. Here, we demonstrated that culturing ETEC strains in an IEC-conditioned medium altered the mRNA expression of T1SS and T2SS components. In the presence of epithelial factors, the transcript levels of the T1SS components TolC and DsbA mirrored the ST secretion levels in strains GIS26 and IMM10. This seems to indicate that the expression of ST and T1SS are co-regulated. In contrast, the transcript levels of the T2SS components GspD and YghG do not match with the LT secretion levels in response to secreted epithelial factors in the tested ETEC strains. This suggests that the delivery of LT may involve mechanisms on top of the T2SS that coordinate the release of holotoxin from the periplasm of ETEC. 

The CRP–cAMP axis has been shown to control secretion systems and to orchestrate regulating cascades involved in enterotoxin gene expression in response to growth conditions [25,40]. Given that cAMP is an important secondary messenger in regulating LT secretion by ETEC, it is intriguing to speculate that CRP may also provide a competitive advantage by modulating ST secretion by ETEC [27,42]. In contrast to a negative association between LT and CRP transcripts, our study revealed that the upregulation of CRP is linked to the increased transcripts of both ST encoding genes and T1SS components [25,27]. Interestingly, a mild heat treatment abrogated the effect of the IEC-conditioned medium on the transcription of some of the investigated genes in ETEC. Heat treatment of the IEC-conditioned medium did not reduce the effect of the secreted epithelial factors on the secretion of LT and pSTa. However, the effect of the epithelial factors on STb secretion was abrogated. This might suggest that the secreted epithelial factor involved in *eltB* and *estA* gene transcription is heat stable, while the epithelial factor modulating the *estB* gene transcription is not. Concerning the T1SS (TolC, DsbA) and T2SS (YghG) components, heat treatment abrogated the effect of the IEC-conditioned medium on their transcription, while *gspD* transcription was unaffected. Likewise, the effect of the IEC-conditioned medium on crp transcription was negated by heat treatment. Collectively, the heat treatment of the IEC-conditioned medium was sufficient to negate the effect of epithelial factors on the transcription of enterotoxins, suggesting that the epithelial factors may be heat sensitive proteins or other small molecules.

In this study, host-derived gut epithelial factors were present in the IEC-conditioned medium. However, further research is needed to identify the secreted gut epithelial factors that are associated with the expression of enterotoxins. In vitro observations suggested a potential role for cAMP in the promotion of ETEC adherence to host cells and in modulation of bacterial adhesion genes [16,21]. Moreover, ETEC strains possess the ability to utilize exogenous cAMP [20,26,27]. Hence, it is likely that the IEC-conditioned medium contained cAMP produced by differentiated IPEC-J2 cells, which is then used by ETEC to regulate its gene expression via CRP. Interestingly, some ETEC strains are more sensitive to extracellular cAMP, which might explain the inter-strain differences we observed. 

In conclusion, it appears that factors secreted by intestinal epithelial cells under steady state conditions affect the transcription of genes encoding enterotoxins in porcine ETEC strains. Although further research is needed to identify these factors, these results offer a novel model for bacteria–intestine interactions in which ETEC sense secreted epithelial factors to orchestrate enterotoxin secretion.

## 4. Materials and Methods

### 4.1. Preparation of Intestinal Epithelial Cell-Conditioned Medium

The porcine jejunal cell line IPEC-J2 (intestinal porcine epithelial cells from jejunum) was originally isolated from the jejunum of a neonatal piglet. The culture medium consisted of 1:1 Dulbecco’s Modified Eagle Medium (DMEM)/Ham’s F-12 mixture (Invitrogen, Geel, Belgium) supplemented with 5% fetal bovine serum (Gibco, Belgium), 1% penicillin–streptomycin (Sigma–Aldrich, Hoeilaart, Belgium), 1% insulin–transferrin–selenium (Invitrogen), 2% l-glutamine (Invitrogen), and 5 ng/mL human epidermal growth factor (Invitrogen). IPEC-J2 cells were maintained at 37 °C, 5% CO_2_, and 90% humidity, and passaged when they reached 90% confluence. For the collection of intestinal epithelial cell-conditioned medium (IEC-conditioned medium), IPEC-J2 cells between passages 55 and 62 were seeded onto collagen-coated 6-well transwell inserts at 5 × 10^5^ cells per inserts (pore size 0.4 μm; 4.67 cm^2^; Corning, Amsterdam, The Netherlands) and grown as monolayers until confluency, as previously described [14]. Upon confluency, IPEC-J2 monolayers were cultured in differentiation medium (IPEC-J2 culture medium without FCS, Diff medium) for 48 h before medium was replaced with fresh one every other day until full differentiation. To monitor differentiation, the trans-epithelial electrical resistance (TEER) was measured using a Millicell Electrical resistance system (Millipore-Merck, Hoeilaart, Belgium). Upon reaching a TEER value of 3000 Ω·cm^2^, the IPEC-J2 monolayers were washed twice with phosphate-buffered saline (PBS) and cultured with differentiation medium without antibiotics. At a TEER value of at least 5000 Ω·cm^2^, IPEC-J2 monolayers were considered as differentiated and used to collect IEC-conditioned medium from the apical side of the transwell [14].

### 4.2. Bacterial Strains and Growth Conditions

The bacterial strains used in this study are listed in Table 1. The selection of the ETEC strains in this study were based on the information previously published [12]. Briefly, the IMM07 and IMM96 strains differ in their ability to secrete the LT enterotoxin, although they have the same production ability. The GIS26 and IMM10 strains were included to evaluate pSTa and STb secretion levels. GIS26 has higher pSTa and STb secretion levels than IMM10.

Strains were grown on brain heart infusion (BHI) agar plates from −80 °C glycerol stocks and grown for 18–24 h at 37 °C in Casamino Acids–Yeast Extract (CAYE) medium with 0.25% glucose (*v*/*v*) in a rotary shaker, set at 180 rpm. The amount of overnight fresh bacteria was determined by measuring the optical density (OD) at 650 nm by spectrophotometry (Cytiva, Hoegaarden, Belgium), and adjusted to the same value with CAYE medium. The 1:100 diluted fresh bacteria were grown for 18–24 h at 37 °C in CAYE, IPEC-J2 differentiation medium (without antibiotics), and IEC-conditioned medium, respectively. In order to investigate which factors secreted by intestinal epithelial cells might be involved in the transcriptional modulation of ETEC, the IEC-conditioned medium was treated at 56 °C for 30 min to inactivate potential factors. ETEC was added to the heat-treated medium at the same concentration as above and the cells were incubated for 18–24 h at 37 °C in a rotary shaker, set at 180 rpm.

### 4.3. Isolation of Total Bacterial RNA

The bacteria were harvested and total RNA was isolated using the TRIzol^®^ Max™ Bacterial RNA Isolation kit (Thermo Fisher, Waltham, MA, USA). Prior to their lysis in TRIzol Reagent, bacteria were treated with Max Bacterial Enhancement Reagent following the manufacturer’s guidelines. To remove DNA, total bacterial RNA was treated with RQ1 RNase-Free DNase (Promega, Madison, WI, USA) and purified with the RNeasy Mini Kit (Qiagen Benelux, Venlo, The Netherlands), according to the kit instructions. The RNA concentration and purity were determined with a DS-11 spectrophotometer (DeNovix, Wilmington, DE, USA). Only the samples with OD_260_/OD_280_ ratios between 1.8 and 2.1, and OD_260_/OD_230_ ratios between 1.7 and 2.0 were reverse transcribed into cDNA using Superscript^TM^ III Reverse Transcriptase (Thermo Fisher) and random primers (Thermo Fisher). Briefly, 1 μg of total RNA was converted to cDNA, diluted 8× in nuclease-free water and used to analyze the transcript levels of selected genes by qPCR.

### 4.4. qPCR

To assess transcriptional changes in ETEC induced by secreted epithelial factors, several genes were evaluated by RT-qPCR. Enterotoxin-encoding genes (*eltB*, *estA*, and *estB*), T1SS-associated genes (*dsbA* and *tolC*), T2SS-associated genes (*gspD* and *yghG*), and a key transcriptional regulator (*crp*) were selected. Prior to qPCR, gene-specific primers and reference genes were optimized. Specific primers were designed using BLAST primer design and confirmed by melting curve analysis as previously published [12]. The amplification efficiency of all the reactions ranged from 95% to 105% using serially diluted cDNA. Expression stability of 8 candidate reference genes was analyzed by geNorm software in samples of native CAYE medium, Diff medium, and IEC-conditioned medium, and 3 reference genes, cysG, ropA, and hcaT, with a stable expression were selected as reference genes to perform normalization. Relative gene expression was quantified on a StepOnePlus real-time PCR system (Thermo Fisher) with specific primers in PowerUp SYBR Green PCR Master Mix (Thermo Fisher). Gene-specific transcripts were normalized to the three reference genes using the 2^−ΔΔCt^ method. All reactions were performed in triplicate.

### 4.5. Detection of Enterotoxins Secreted by ETEC Strains Grown in Different Media

To examine secretion of LT, pSTa, and STb by porcine-specific ETEC strains in CAYE, Diff, IEC-conditioned medium, and heat-treated IEC-conditioned medium, bacterial culture supernatants and periplasmic extracts were collected. LT, pSTa, and STb levels were assessed by GM1-ELISA, a competitive ELISA, and Western blotting, respectively, as described [12].

### 4.6. Data Analysis

The experiments were performed in triplicate on at least three separate occasions. The concentration of secreted LT and pSTa were determined with DeltaSoft JV software (Version 2.1.2, BioMetallics, Princeton Junction, NJ, USA). Quantification of STb band intensities was performed using ImageJ. Statistical analysis was performed using SPSS 26. The transcribed mRNA levels were analyzed using qBaseplus software (Biogazelle, Ghent, Belgium) and analyzed by a non-parametric Kruskal–Wallis analysis, followed by a Dunn’s post hoc test. A *p*-value < 0.05 was considered statistically significant.

## Figures and Tables

**Figure 1 ijms-23-06589-f001:**
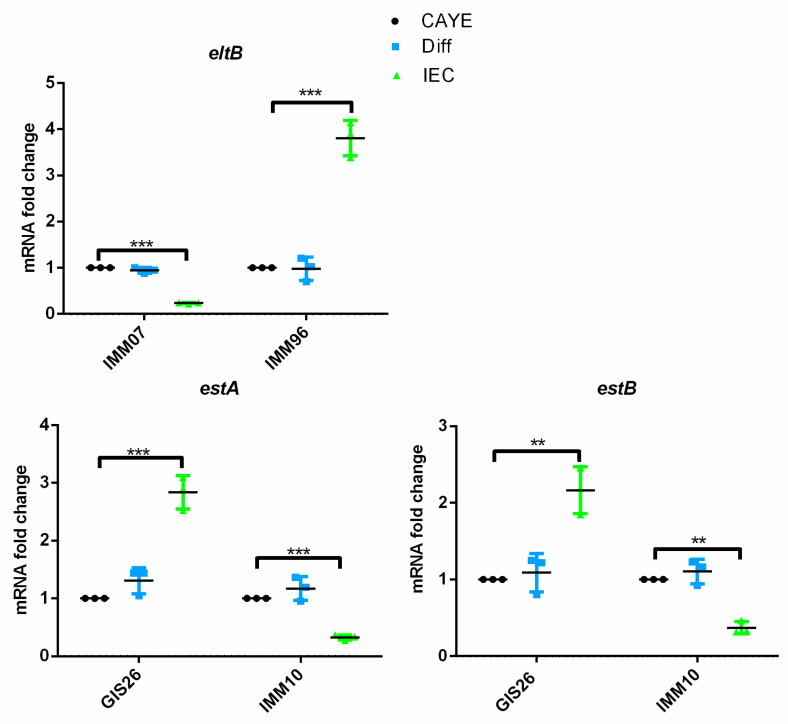
Factors secreted by gut epithelial cells significantly altered transcript levels of enterotoxins in ETEC strains. The mRNA expression levels of enterotoxins were assessed by qPCR in ETEC strains. These strains were cultured in CAYE medium, IPEC-J2 differentiation medium (Diff), or apical medium obtained from intestinal epithelial monolayers (IEC). The mRNA expression level of the targeted genes was normalized to three reference genes and presented as the fold change to CAYE medium. Data are presented as the mean ± SD. ** *p* < 0.01, *** *p* < 0.001.

**Figure 2 ijms-23-06589-f002:**
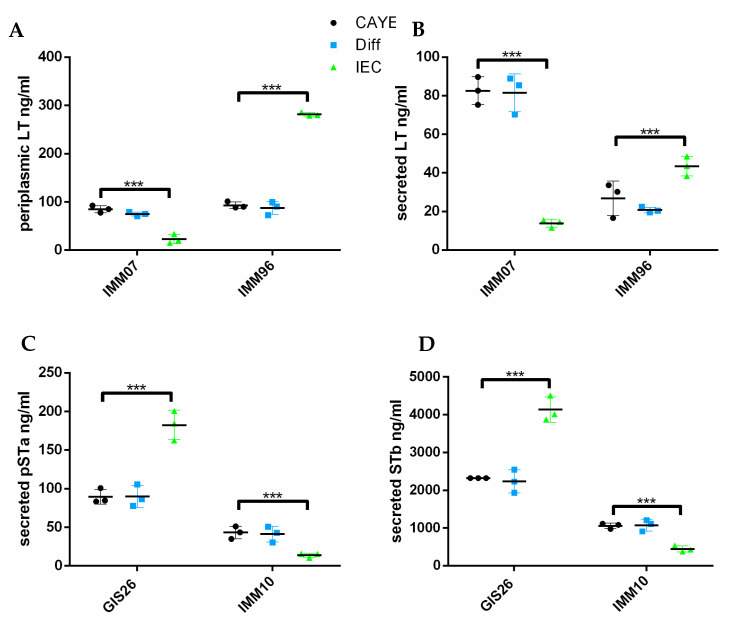
Porcine strains differ in their ability to secrete enterotoxins in response to secreted epithelial factors. ETEC strains were grown in CAYE medium, IPEC-J2 differentiation medium (Diff), and IEC-conditioned medium. Periplasmic LT production (**A**) and secretion levels (**B**) were quantified by GM1-ELISA. (**C**) The pSTa secretion levels were quantified by an in-house competitive ELISA. (**D**) The secreted STb levels expressed by ETEC strains were evaluated by Western blotting and quantified by ImageJ. Data are presented as the mean ± SD. *** *p* < 0.001.

**Figure 3 ijms-23-06589-f003:**
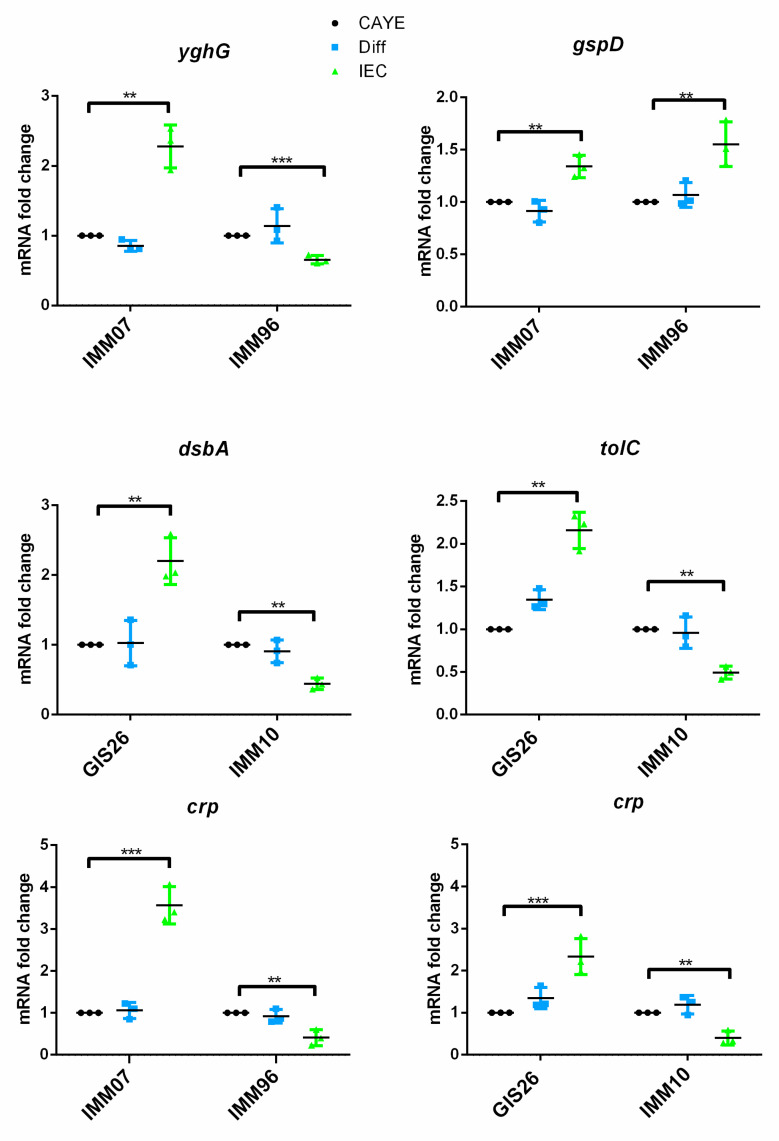
The mRNA expression levels of components of T2SS and T1SS and the key transcriptional regulator CRP. gspD and yghG genes were selected as key components of the T2SS system, while tolC and dsbA genes were selected as key components of the T1SS system. Transcript levels were assessed by qPCR in the ETEC strains. The mRNA expression level was normalized to three reference genes. Data are presented as the mean ± SD. ** *p* < 0.01, *** *p* < 0.001.

**Figure 4 ijms-23-06589-f004:**
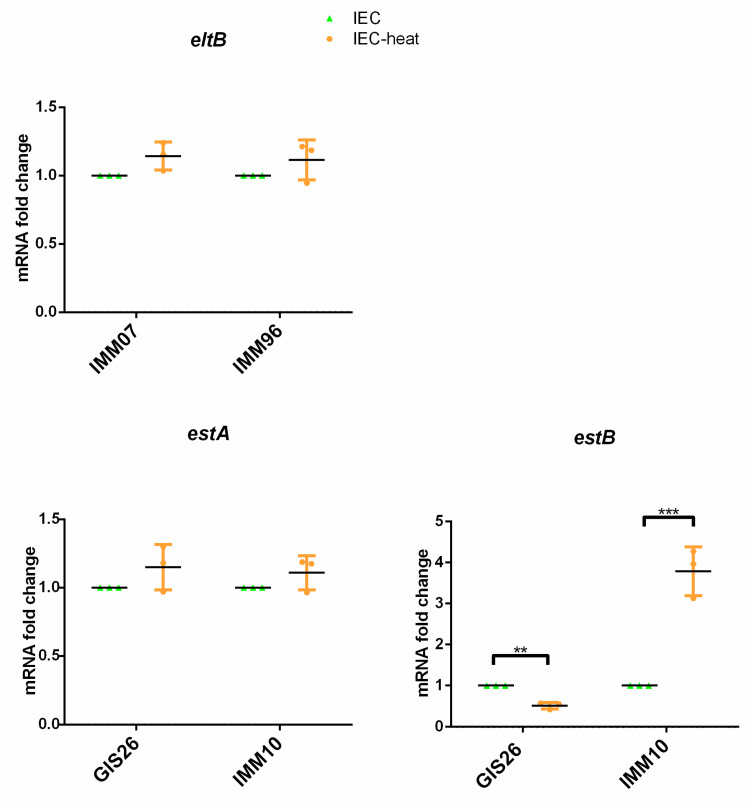
The mRNA expression levels of enterotoxin genes, crucial components of T1SS, T2SS, and the key transcriptional regulator CRP, were assessed by qPCR in ETEC strains grown in IEC- and heat-treated IEC-conditioned medium. The mRNA expression level was normalized to three reference genes. Data are presented as the mean ± SD. * *p* < 0.05, ** *p* < 0.01, *** *p* < 0.001.

**Figure 5 ijms-23-06589-f005:**
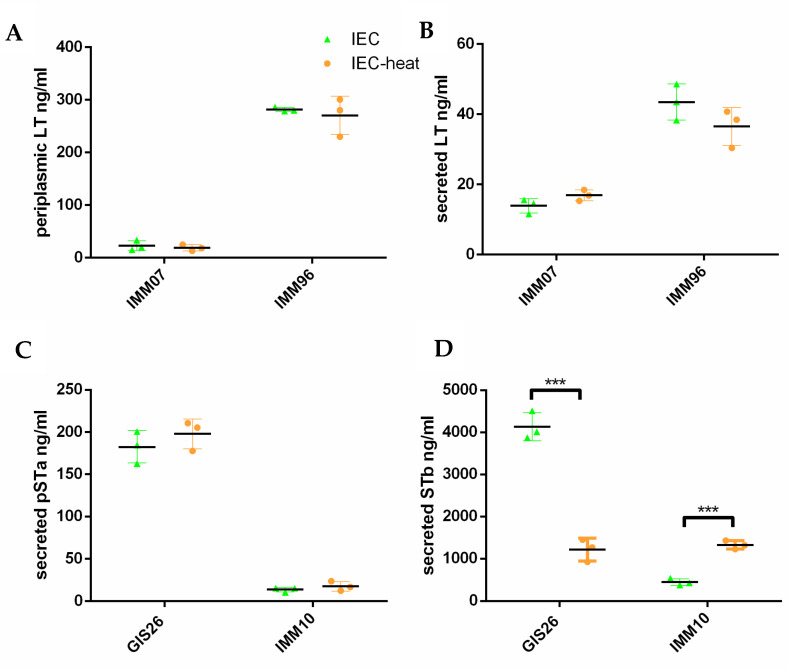
Heat treatment of media conditioned by epithelial cells (IEC)-affected secretion of enterotoxins. ETEC strains were grown in IEC-conditioned medium (IEC) or heat-treated IEC-conditioned medium (IEC-heat). Periplasmic LT production (**A**) and secretion levels (**B**) were quantified by GM1-ELISA. (**C**) The pSTa secretion levels were quantified by an in-house competitive ELISA. (**D**) The secreted STb levels expressed by ETEC strains were quantified by ImageJ. Data are presented as the mean ± SD. *** *p* < 0.001.

**Table 1 ijms-23-06589-t001:** The bacterial strains used in this study.

Strain Name	Enterotoxins	Infection
GIS26	LT^+^pSTa^+^STb^+^	PW
IMM07	LT^+^pSTa^−^STb^+^	PW
IMM10	LT^+^pSTa^+^STb^+^	N
IMM96	LT^+^pSTa^−^STb^+^	PW

PW: postweaning, N: neonatal.

## Data Availability

Data reported in this study are available upon request.

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
