# Peer review of "Intestinal Epithelial Cells Modulate the Production of Enterotoxins by Porcine Enterotoxigenic E. coli Strains"

_ijms, 2022, doi:10.3390/ijms23126589_

Round 1

Reviewer 1 Report

The authors present the results of their study aimed at investigating the effects of polarized IPEC-J2 cells-derived factors on ETEC-derived virulence factors. Specifically, the authors looked at genes encoding enterotoxins, secretion system associated proteins, and the regulatory molecules cyclic AMP (cAMP) and receptor protein (CRP). The study provides new and interesting results that are of value to this field of research.

The study is well planned and the results are presented in a clear form. The manuscript is generally well written. I do however have a few comments:

(1) It is not clear to me why the authors shift in their presentation of the heat-stable enterotoxin a, but not for heat-stable enterotoxin b. In some places (line 46) STa is used as pSTa and in other places (line 94) it is used as hSTa. There are no definitions for either. Perhaps the authors can simply use STa for all mentions, regardless of the source of the toxin.

(2) Lines 93-94: The reference used at the end of that statement (reference 25) did not report results on STa expression, only on LT.

(3) When referring to a figure in the text, please capitalize the “f”. Line 128 as an example: Figure 1 instead of figure 1.

(4) Lines 141-142: LT is secreted through the outer membrane via GspD but in lines 65-66 the statement implies that TolC is involved in the secretion of LT. Please confirm or correct.

(5) Line 150: The statement is incorrect. STa and STb secretion is not a one-step process but involves the periplasmic disulfide isomerase enzyme DsbA. Both toxins have disulfide bonds and transit through the periplasm before secretion through the outer membrane. It is a two-step process with passage through the SecYEG translocon and signal sequence cleavage, followed by formation of disulfide bonds and export through the TolC within the outer membrane.

(6) Line 151: The statement of the involvement of AcrAB in the export of STb is not referenced. What is known about this step is the involvement of MacAB, and not AcrAB [ Yamanaka, H., Kobayashi, H., Takahashi, E., & Okamoto, K. (2008). MacAB is involved in the secretion of Escherichia coli heat-stable enterotoxin II. Journal of bacteriology, 190(23), 7693-7698].

Reviewer 2 Report

The paper ‘Intestinal epithelial cells modulate the production of enterotoxins by porcine enterotoxigenic E. coli strains’ by Wang H, et al., investigated on the sensitivity and response of ETEC strains to molecules secreted by intestinal epithelial cells. The presence of molecules secreted by differentiated-epithelial monolayers altered enterotoxin-gene transcription in ETEC in a strain specific manner. The results suggested that host-derived factors may take part in the regulation of virulent factors in ETEC. The study is well planned, and the manuscript is well written. Although further research is needed to identify the epithelial secreted factors, the results offer a novel mechanism for ETEC-sensing, and bacteria-intestine interactions.

Author Response

We would like to take the opportunity to thank reviewer to read our manuscript and provide such positive comments.